# A Comparative Evaluation of Gait between Children with Autism and Typically Developing Matched Controls

**DOI:** 10.3390/medsci5010001

**Published:** 2017-01-06

**Authors:** Janet S. Dufek, Jeffrey D. Eggleston, John R. Harry, Robbin A. Hickman

**Affiliations:** 1Department of Kinesiology and Nutrition Sciences, University of Nevada, Las Vegas, NV, 89154, USA; egglest6@unlv.nevada.edu (J.D.E.); harry@unlv.nevada.edu (J.R.H.); 2Department of Physical Therapy, University of Nevada, Las Vegas, NV, 89154, USA; robbinhickman@gmail.com

**Keywords:** autism spectrum disorder, gait, matched-pair analysis, rehabilitation, walking

## Abstract

Anecdotal reports suggest children with autism spectrum disorder (ASD) ambulate differently than peers with typical development (TD). Little empirical evidence supports these reports. Children with ASD exhibit delayed motor skills, and it is important to determine whether or not motor movement deficits exist during walking. The purpose of the study was to perform a comprehensive lower-extremity gait analysis between children (aged 5–12 years) with ASD and age- and gender-matched-samples with TD. Gait parameters were normalized to 101 data points and the gait cycle was divided into seven sub-phases. The Model Statistic procedure was used to test for statistical significance between matched-pairs throughout the entire gait cycle for each parameter. When collapsed across all participants, children with ASD exhibited large numbers of significant differences (*p* < 0.05) throughout the gait cycle in hip, knee, and ankle joint positions as well as vertical and anterior/posterior ground reaction forces. Children with ASD exhibited unique differences throughout the gait cycle, which supports current literature on the heterogeneity of the disorder. The present work supports recent findings that motor movement differences may be a core symptom of ASD. Thus, individuals may benefit from therapeutic movement interventions that follow precision medicine guidelines by accounting for individual characteristics, given the unique movement differences observed.

## 1. Introduction

Autism spectrum disorder (ASD) affects an individual’s life and is associated with impaired social interactions and stereotypy [1]. The prevalence of ASD has continually risen in recent years with current prevalence estimations ranging from one in 45 [2] to one in 68 [1]. Children can benefit from early diagnoses at younger ages, taking advantage of sensitive periods in which rapid age-related changes occur in the developmental domains most strongly identified with autism: social behaviors, cognition, and communication [3]. Although the incident rate of ASD continues to rise, clinicians struggle to accurately diagnose children with the disorder [4]. These difficulties may be related in part to limitations in the current diagnostic standards in the Diagnostic-Statistics Manual (DSM-5) [5] in which behavioral characteristics, such as, abnormal social reciprocity, highly restrictive interests, or hyper- or hypo-reactivity to sensory stimuli, are emphasized [6]. Unfortunately, the behavioral emphasis during subjective clinical observations and treatments has done little to bring positive changes to individuals affected by ASD [7].

In the 1990s, Teitelbaum and colleagues suggested that movement analyses could be beneficial for diagnosing ASD during infancy [8]. However, the current age threshold for diagnoses is two years of age and the typical age for diagnosis is four years of age [9]. Delayed diagnoses may be related to the small repertoire of observable social and communication skills at such a young age. Contemporary research suggests that ASD is not solely a mental health disorder, and that it may also present clinically as a movement disorder [10,11] characterized by different manifestations displayed across body systems and developmental domains [12]. Movement challenges associated with ASD include delayed motor milestone development [13], altered muscle recruitment patterns [14,15], impaired postural control [15], and deficits in locomotion and balance [12] and dyspraxia [12,13,16,17,18]. These deficits present functional motor challenges to children with ASD and may influence other areas of development. Theories of grounded cognition [19] and language acquisition [20] suggest that fine and gross motor skills necessary for environmental exploration and object manipulation may open the door for language and social behaviors to emerge.

Although motor delay and dysfunction is a component of ASD, motor behaviors are not included in the DSM-5 diagnostic criteria or the International Classification of Functioning, Disability and Health (ICF) core set for ASD [21]. Further, recognizable or predictable patterns of motor behaviors have not been established. Thus, it is important to recognize gross motor movement limitations clinically present in children with ASD during routine movement tasks like walking. Neuroimaging studies offer evidence that elements of the perception-action continuum, including the primary motor cortex [22], basal ganglia [23] and cerebellum [24], are different in children with ASD than their peers. However, clinical evidence quantifying how these differences manifest themselves in important motor behaviors such as walking remains inconclusive. Reasons for these inconsistencies can include factors such as variations in data collection protocols and sample sizes. However, the heterogeneous nature of ASD may better explain the lack of conclusive findings.

There is a paucity of scientific literature in which simultaneous kinetic and kinematic gait parameters have been used to examine walking in individuals with ASD [25]. Previous studies identified differences in cadence and peak hip and ankle flexion and torque [25] as well as differences in ankle range of motion [26] between individuals with ASD compared to individuals with typical development (TD). However, to the authors’ knowledge, no study has examined lower extremity gait parameters performed across a complete gait cycle by using a point-to-point analysis. A comprehensive analysis of the entire movement pattern may provide important information relative to force attenuation and segmental control capabilities in this population, thus uncovering distinct movement characteristics among children with ASD. Broadening our understanding of movement patterns in children with ASD may support the inclusion of movement during diagnoses and help to present intervention strategies to promote motor skill acquisition and minimize movement deficiencies [27,28]. An important consideration during comparative analyses in this population is the statistical approach. Because group analyses hinge upon the assumption of homogeneity, the unique characteristics exhibited by each individual are masked and typically regress to the mean [29,30]. The consequences of this masking effect warrant the use of matched-pair analyses that compare one child with ASD to one child with TD, which can provide a more complete interpretation of the heterogeneity of the population. The purpose of this study was to perform a bilateral, matched-pair examination of lower extremity gait parameters between children with ASD and matched samples with TD used as a comparative norm. Due to the distinct neurological manifestations of ASD [23,27,31], we expected similar observations with respect to motor function. As such, we hypothesized that statistically significant differences, unique in terms of directionality of differences (increased vs. decreased flexion/force), would be revealed across the gait cycle for each matched-pair with respect to hip, knee, and ankle joint angles and vertical and anterior-posterior ground reaction forces (vGRF and AP GRF, respectively).

## 2. Materials and Methods

### 2.1. Participants

Ten children with ASD and ten children with TD aged between 5 and 12 years participated in the study (14 males, 6 females; 9.0 ± 2.2 years 1.4 ± 0.2 m, and 34.8 ± 13.4 kg, ASD; and 9.0 ± 2.1 years, 1.4 ± 0.1 m, and 35.7 ± 10.2 kg, TD). Because we performed a matched-pair analysis (one child with ASD compared to one age-gender matched child with TD), the total number of participants was trivial in comparison to the number of trials evaluated [32]. Accordingly, sample size was based on the fact that we evaluated 20 trials for each matched-pair analysis. Nevertheless, we performed a traditional a priori power analysis to determine the necessary sample size to achieve adequate statistical power. We performed this power analysis using ankle range of motion data of Hallet et al. [26] with a proposed effect size of 1.5, power (1-beta) of 0.8, and a significance level (alpha) of 0.05. A total sample of 16 participants was suggested, with equal numbers of children with ASD and children with TD. To participate in the study, children with ASD were required to present proof of a clinical diagnosis of the disorder from a medical professional that was verbally confirmed by each child’s parent. We did not attempt to classify children according to ASD levels of severity, as we did set out to correlate functional level to the severity of gait differences between children with ASD and peers with TD and wished to reflect the natural heterogeneity of this population. Children with TD were required to not have a clinical diagnosis of ASD and to also be age- and gender-matched to a study-enrolled child with an ASD diagnosis. We chose to age- and gender-match these pairs independent of cognitive level since deficits in motor ability in children with less severe ASD do not reliably correlate with cognitive ability at the individual level [33]. Due to the heterogeneity of the manifestation of ASD, children with ASD were not excluded due to varied gait patterns (i.e., toe walking). This study was approved by the Institutional Review Board (#710824) at the host institution (University of Nevada, Las Vegas) for use at the site institution (Boise State University) where data collection occurred. Parental consent and child assent were obtained prior to participation.

### 2.2. Instrumentation

An eight-camera motion capture system (120 Hz; Vicon Motion Systems, Oxford, UK) tracked retro-reflective marker trajectories placed bilaterally on participants’ lower-extremities. One Kistler (480 Hz; Kistler Instrument Corp., Amherst, NY, USA) and two AMTI (480 Hz; Advanced Mechanical Technology Inc., Watertown, MA, USA) force platforms were mounted flush with the floor, and were used to obtain three-dimensional ground reaction force (GRF) data bilaterally during walking trials.

### 2.3. Procedures

Nineteen retro-reflective markers were placed bilaterally at the following locations: base of the second and fifth toes, calcaneus, medial and lateral malleoli, medial and lateral knee joint line, anterior superior iliac spines, and posterior superior iliac spines, with an additional marker placed on sacrum. Three-marker clusters were placed bilaterally on the lateral aspect, mid-segment, on the thighs and legs. Participants were instructed to stand in a “T” pose for a static calibration trial that was used to establish zero degrees of joint flexion. Participants were instructed to perform 20 walking trials at a self-selected pace. Participants were instructed to walk as normally as possible, and were reminded to do so when necessary. Walking velocity was not directly controlled unless a participant walked at an uncharacteristically slow, fast, or uneven pace. The children with ASD exhibited an average walking velocity of 1.27 ± 0.22 m/s, while the children with TD exhibited an average walking velocity of 1.30 ± 0.18 m/s.

### 2.4. Data Reduction

Walking trials were reduced to strides where participants had at least one complete foot strike on one of the three force platforms. The average number of left and right strides obtained from the 20 walking trials was 22.5 ± 5.51, ranging between 13 and 34 strides. Raw data were exported to the Visual 3D biomechanical software suite to filter marker trajectories and GRF data with a low-pass Butterworth digital filter (6 Hz and 25 Hz, respectively). Sagittal plane hip, knee, and ankle joint positions were computed bilaterally. GRF data were examined in the vertical and anterior-posterior axes, and were normalized to multiples of participants’ body weight. All data were normalized to 100% of the gait cycle (101 data points) and exported to Matlab (The Mathworks, Inc., Natick, MA, USA) for statistical analyses.

### 2.5. Statistical Analysis

To evaluate the complete gait cycle, we used a contemporary data analysis procedure [34] that statistically compares each of the 101 data points per variable. To perform this analysis, ensemble mean-time and standard deviation-time histories were first computed bilaterally across strides per participant for vGRF, AP GRF, and sagittal plane hip, knee, and ankle joint positions. To test for statistical significance, Model Statistic analyses [32,34,35,36] were conducted at each of the 101 data points across the gait cycle [34] between the ensemble mean-time histories of the matched pairs. Although the Model Statistic was designed as a single-subject statistical procedure, we performed this procedure on a matched-pair basis to account for any within subject variability that might have influenced movement pattern differences between children with ASD and children with TD. The gait cycle was divided into the following sub-phases as a percent range of the gait cycle [37]: initial contact and loading response (0%–10%), mid-stance (11%–30%), terminal stance (31%–50%), pre-swing (51%–60%), initial swing (61%–73%), mid-swing (74%–87%), and terminal swing (88%–100%). The number of significant differences across sub-phases between each matched pair, and collapsed across all subjects, was presented as a percentage of the total number of data points throughout each sub-phase and the total gait cycle.

## 3. Results

### 3.1. Ground Reaction Forces

The differences in vGRF and AP GRF were examined across the stance phase. For vGRF, an average of 34.7 ± 13.5 (right limb) and 42.7 ± 7.3 (left limb) significant differences (*p* < 0.05) out of a possible 60 significant differences were observed between the children with TD and ASD across the 10 matched pairs. For AP GRF, an average of 26.6 ± 13.2 (right limb) and 32.4 ± 15.6 (left limb) significant differences (*p* < 0.05) were observed between the children with TD and ASD across the 10 matched pairs.

The following values represent the percentages of significant differences (*p* < 0.05) in vGRF across all 10 pairs during the sub-phases of stance phase. During the loading response, 66.0% ± 28.7% (right limb) and 100% (left limb) was significantly different. During mid-stance, 66.0% ± 32.6% (right limb) and 50.6% ± 32.4% (left limb) was significantly different. During terminal stance, 48.5% ± 31.5% (right limb) and 88.7% ± 21.0% (left limb) was significantly different. During pre-swing, 52.0% ± 38.8% (right limb) and 48.7% ± 42.9% (left limb) of the phase was significantly different. For AP GRF, differences were observed during 57.0% ± 38.6% (right limb) and 100% (left limb) of loading response, 47.0% ± 35.7% (right limb) and 100% (left limb) of mid-stance, 39.5% ± 37.3% (right limb) and 43.7% ± 33.9% (left limb) of terminal stance, and 36.0% ± 29.1% (right limb) and 38.7% ± 28.4% (left limb) of pre-swing (*p* < 0.05).

### 3.2. Joint Angles

Of the 101 data points comparisons, the hip exhibited 87.6 ± 8.7 (right limb) and 90.7 ± 4.1 (left limb) differences (*p* < 0.05). The knee exhibited 69.1 ± 15.0 and 81.1 ± 12.42 differences (*p* < 0.05). The ankle exhibited 73.1 ± 16.4 (right limb) and 82.0 ± 14.8 (left limb) differences (*p* < 0.05). An exemplar representation of the number of observed sagittal plane joint angle differences between one matched-pair of children with TD and ASD is presented in Figure 1. The percentages of significant differences (collapsed across matched pairs) for the right and left hip, knee, and ankle joints during each sub phase are presented in Figure 2. Additionally, the percentages of significant differences during each of the gait sub-phases for each individual matched pair at each joint are presented in Table 1. Although many differences were identified between each matched-pair at each joint examined, there was little consistency with respect to the presence of greater/lesser flexion/dorsiflexion in the children with ASD compared to the children with TD throughout specific sub-phases of the gait cycle. An exemplar representation of the inconsistent joint angle differences is presented in Figure 3.

## 4. Discussion

The purpose of this study was to compare gait characteristics between children with ASD compared to age- and gender-matched peers with TD. Due to the heterogeneity of ASD, a matched-pair statistical approach was used to identify distinct individual differences that have yet to be identified using traditional statistical approaches. In support of our hypothesis, numerous significant differences were observed in the vGRF, AP GRF, and sagittal plane time-histories of the hip, knee, and ankle for each of the 10 matched pairs. Although many significant differences were detected among the lower extremity joint angles, no patterns were identified relative to the directionality of the observed differences. This finding is consistent with the literature [10,13,27] documenting the heterogeneous nature of ASD, and supports the fact that no two individuals with the disorder are the same [38]. It should be noted that although these children with ASD exhibited gross motor differences compared to their age- and gender-matched peers with TD, a causal relationship between motor dysfunction and impaired behavioral and communication skills has yet to be established.

Specific to the vGRF data, the differences identified within each pair combined with the lack of observable patterns across the matched-pairs indicate that children with ASD use distinct strategies to attenuate impact and load the body in comparison to their peers with TD. Additionally, the differences and characteristics of the AP GRF profiles support previous research indicating that children with ASD lack stability and possess muscular weakness in comparison to children with TD [27]. Decreased stability and strength may present greater challenges to children with ASD during the transition between single limb support and double limb support during walking. The loading response is a mechanism during walking that utilizes the body’s dampening systems (i.e., plantar fascia in the foot, ankle range of motion) to reduce the magnitude of impact force experienced by the bones and tissues [39]. The initial actions of the loading response are to plantarflex the ankle joint after ground contact and use the available range of motion in combination with the soft tissues of the foot to dampen the forces at impact. A potential implication of dysfunctional loading is that the impact force from ground contact may be repeatedly redirected to a structure that is less capable of attenuating these forces, leading to potential damage to the musculo-skeletal system.

The distinct loading and movement pattern strategies observed in these data appear to be the result of dysfunctional lower extremity positioning and/or control specific to each child. It is likely that the distinct strategies reflect inconsistent or unpredictable movement responses to the combination of different sources of peripheral noise [38] and different sensory capabilities [40] affecting proprioception when the foot contacts the ground. Loading the body differently over numerous task repetitions may offer some advantages, since variable loading can mitigate the repetitive stress experienced by bony and soft tissues [41]. However, a potential consequence of the inconsistent movement patterns identified in these children with ASD is in their ability to rapidly and appropriately adapt their movements to environmental and/or task changes. If a child with ASD cannot consistently move his/her foot when approaching and contacting the ground during repetitive walking actions, the risk of acute injury related to a trip or a fall might increase, in addition to the potential for overuse injury related to musculo-skeletal overload [42]. The potential for injury might further increase when faced with a movement challenge, such as stepping up onto a curb or avoiding an object on the ground, as the sequential processes that contribute to position the foot and elevate the body up onto the curb may be more difficult and delayed for a child with ASD.

Interestingly, we observed consistently fewer differences across matched-pairs in the knee and ankle angle data compared to the hip angle data when averaged across sub-phases and pairs (Table 1). Although these participants were all experienced walkers, we believe that the children with ASD constrained their range of motion at the knee and ankle during the loading response as a protective mechanism to keep from falling, similar to the strategy used while learning a new motor skill [43]. Constrained degrees of freedom are released in a proximal to distal pattern [44,45] which supports our conclusion that these children with ASD constrained their ankle motion and released the constrained motion at the proximal joints, greater numbers of significant differences, to produce forward motion and maintain adequate stability. Although children may not develop a fully mature gait pattern until around the age of seven, it is possible that this proposed protective mechanism may be a characteristic of the movement in children with ASD. This result is similar to what is observed in individuals who are post-stroke [46]. The loss of stability and reduced motion at the ankle joint is likely due to motor control [14,15] and praxis issues [17,18] observed in children with ASD.

Gross motor movements are generally performed in the sagittal plane, and therefore, we believe the differences observed in these data represent distinct gross motor dysfunction in this sample of children with ASD. Large standard deviation values observed among our sample of children with ASD (Figure 2) allude to a larger magnitudes of joint motion variability compared to that seen in children with TD. These differences indicate that children with ASD exhibit decreased lower body control and limited movement repeatability compared to children with TD. These deficits may be the source of the inconsistent loading mechanisms described previously. Kindregan et al. [27] suggested that obtaining a more thorough understanding of gross movement dysfunction in children with ASD might increase the referral rate to therapeutic interventionists aimed to mitigate deficits in muscular weakness, postural and balance control. Therapeutic interventions that aim to address the movement deficits identified herein could improve the ability of children with ASD to functionally ambulate more consistently and efficiently within a constantly changing environment [28].

### Study Limitations

A limitation of the current study was the inability to independently verify each child’s ASD diagnosis. We addressed this limitation by requiring proof of medical diagnosis and asking the parent(s) of each child with ASD to verbally confirm their child’s diagnosis. The working assumption in the current study was that the children with TD exhibit desirable, mature, or “normal” gait patterns. Any deviations from this assumption would limit the study results. The gait patterns exhibited by these children with TD did not exhibit any overt deviations with gross clinical observation. Although we did not pair based on cognitive level according to Manjiviona and Prior [33], we acknowledge that this may be a limitation as other studies suggest cognitive level is a necessary control in this population. Finally, walking velocity was not directly controlled due to the heterogeneity of walking mechanics in children with ASD. To address this limitation, trials were discarded if a child walked uncharacteristically fast or slow.

## 5. Conclusions

This analysis supports previous findings that ASD is a pervasive disorder that results in differences of function across affected individuals, and contributes to heterogeneous characterization of their movement patterns [13,25,26,28]. These results indicate that the children with ASD exhibited unique variations in impact force attenuation and segmental control strategies during walking compared to their matched peers. In light of these outcomes, we expect that children with ASD will benefit from a combination of social-behavioral therapy and precision medicine aimed to address deficits in stability and movement repeatability, which could mitigate risks for potential injuries associated with trips or falls [27,28]. The distinct movement characteristics indicate that children with ASD require individualized clinical observations specific to motor control strategies and potential impairments to function that can contribute to atypical control strategies. Children with ASD can perform similar motor tasks as their peers, but in a less uniform and more complex manner. Future research emphasizing movement quality during functional tasks (load carriage, stair descent, etc.) and their effects is warranted.

## Figures and Tables

**Figure 1 medsci-05-00001-f001:**
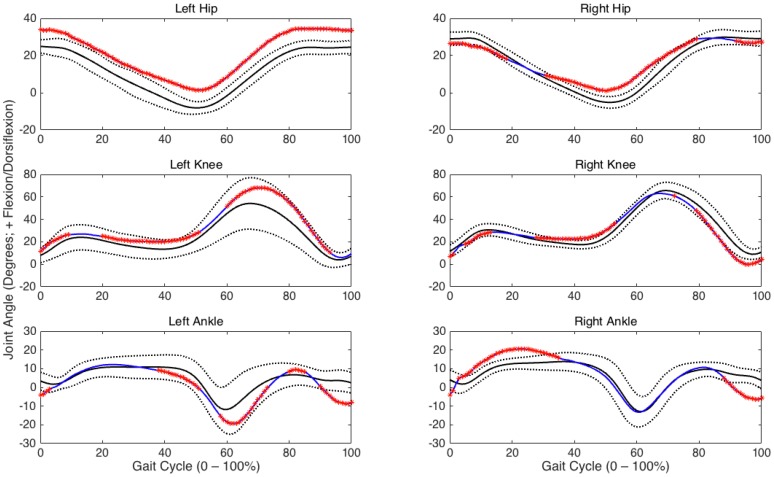
Differences in average hip (**top row**), knee (**middle row**), and ankle (**bottom row**) joint angles between an exemplar matched-pair of typical development (TD) and autism spectrum disorder (ASD, non-toe walker) children (Pair 10) throughout one gait cycle. Note: 0% represents ground contact; 100% represents the end of swing; Mean joint angles for the child with TD are represented by the solid black line; Mean joint angles for the child with ASD are represented by the solid blue line; The area between the grey dashed lines represents a ± 2 standard deviation band associated with the mean joint angles of the child with TD. Significant data point differences (*p* < 0.05) for ASD compared to TD are denoted on the ASD curve with a red asterisk.

**Figure 2 medsci-05-00001-f002:**
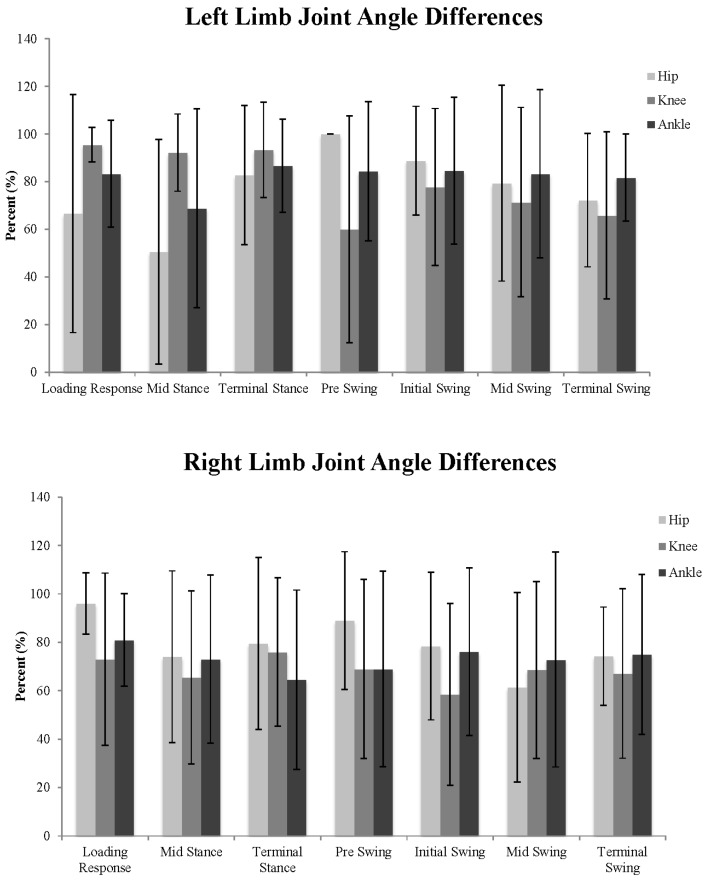
Mean ± one standard deviation of the number of significant sagittal plane joint angle differences (expressed as a percentage of the gait cycle sub-phase) between TD and ASD children averaged across the 10 matched pairs for the left and right hip, knee, and ankle joints.

**Figure 3 medsci-05-00001-f003:**
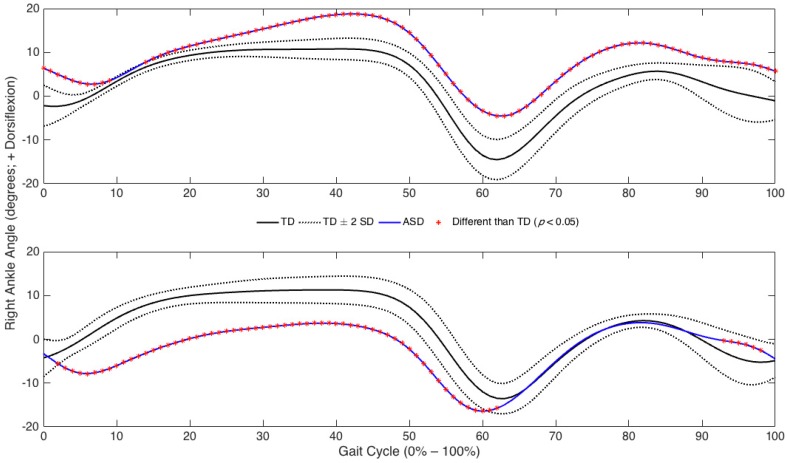
Differences in average right ankle angles over one gait cycle between exemplar matched-pairs of TD and ASD children (Pairs 1 & 3). Note: 0% represents ground contact; 100% represents the end of swing; Mean joint angles for the child with TD are represented by the solid black line; Mean joint angles for the child with ASD are represented by the solid blue line; The area between the grey dashed lines represents a ± 2 standard deviation band associated with the mean joint angles of the child with TD. Significant data point differences (*p* < 0.05) for ASD compared to TD are denoted on the ASD curve with a red asterisk.

**Table 1 medsci-05-00001-t001:** Percentages of significant differences observed for hip, knee, and ankle position by Gait Phase, Limb and Matched Participant Pair.

**Hip Position**	**Stance Sub-Phases**				**Swing Sub-Phases**				**Complete Gait Cycle**	
**Matched Pair**	**Left**		**Right**		**Left**		**Right**		**Left**	**Right**
	**Mean**	**SD**	**Mean**	**SD**	**Mean**	**SD**	**Mean**	**SD**	**Mean**	**Mean**
1	75%	35%	29%	37%	52%	16%	63%	14%	58%	42%
2	x	x	89%	11%	x	x	90%	39%	x	90%
3	100%	4%	100%	4%	98%	48%	98%	48%	100%	100%
4	100%	4%	85%	27%	98%	48%	73%	26%	100%	76%
5	36%	43%	83%	19%	63%	11%	77%	30%	48%	77%
6	54%	41%	100%	46%	36%	8%	67%	23%	45%	83%
7	33%	18%	76%	32%	74%	24%	66%	20%	52%	71%
8	78%	4%	100%	4%	93%	41%	98%	48%	82%	100%
9	100%	4%	100%	47%	98%	48%	41%	27%	100%	69%
10	100%	4%	85%	27%	98%	48%	73%	26%	100%	76%
**Knee Position**	**Stance Sub-Phases**				**Swing Sub-Phases**				**Complete Gait Cycle**	
**Matched Pair**	**Left**		**Right**		**Left**		**Right**		**Left**	**Right**
	**Mean**	**SD**	**Mean**	**SD**	**Mean**	**SD**	**Mean**	**SD**	**Mean**	**Mean**
1	98%	4%	39%	26%	98%	47%	82%	25%	99%	60%
2	x	x	90%	23%	x	x	75%	31%	x	83%
3	100%	4%	100%	39%	98%	48%	74%	27%	100%	90%
4	63%	41%	68%	32%	87%	20%	92%	26%	75%	79%
5	75%	41%	69%	0%	28%	21%	0%	35%	61%	47%
6	100%	34%	56%	21%	67%	29%	83%	26%	87%	66%
7	100%	42%	100%	37%	55%	25%	53%	28%	74%	82%
8	89%	32%	65%	30%	60%	25%	42%	15%	78%	49%
9	75%	52%	63%	15%	69%	12%	93%	33%	79%	76%
10	69%	39%	60%	35%	83%	19%	53%	11%	77%	59%
**Ankle Position**	**Stance Sub-Phases**				**Swing Sub-Phases**				**Complete Gait Cycle**	
**Matched Pair**	**Left**		**Right**		**Left**		**Right**		**Left**	**Right**
	**Mean**	**SD**	**Mean**	**SD**	**Mean**	**SD**	**Mean**	**SD**	**Mean**	**Mean**
1	65%	34%	95%	4%	40%	28%	98%	44%	96%	54%
2	51%	4%	x	x	98%	46%	x	x	0%	64%
3	98%	48%	95%	43%	18%	33%	22%	31%	67%	66%
4	50%	24%	81%	4%	98%	34%	98%	41%	88%	68%
5	75%	24%	84%	28%	80%	27%	77%	26%	80%	76%
6	85%	29%	100%	4%	98%	33%	98%	47%	100%	94%
7	78%	44%	73%	39%	98%	23%	98%	26%	86%	91%
8	95%	11%	96%	18%	93%	41%	88%	38%	93%	94%
9	68%	4%	63%	4%	98%	40%	98%	40%	74%	77%
10	55%	43%	41%	24%	29%	12%	74%	21%	54%	47%

Percentage of significant hip (top), knee, (middle), and ankle (bottom) joint position differences between each matched pair of TD and ASD children averaged across the stance sub-phases, swing sub-phases, and the complete gait cycle; Percentages were calculated as the ratio of the number of significant data-point differences and the total number of possible data point differences for each sub-phase of gait; A higher percentage value represents a greater number of significant differences (*p* < 0.05) between paired individuals; Note that “x” represents missing data for one of the paired participants.

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
