# Peer review of "A Comparative Evaluation of Gait between Children with Autism and Typically Developing Matched Controls"

_medsci, 2017, doi:10.3390/medsci5010001_

Round 1

Reviewer 1 Report

The purpose of this study was to describe gait differences between typically developing children and children diagnosed with autism spectrum disorders (ASD). 

The manuscript is very well written and the results are clearly presented and explained. However, I have a significant methodological issue. You did pair your groups by age and gender. However, you should have paired them by cognitive level too. Differences in cognitive levels could truly influence your results. Even in posture studies, this has been the case.

In addition, you don't describe in detail your group with ASD. Where they high functioning children? or did they vary across the spectrum.

 Also the fact that you did not verify diagnoss, actually testing the children and relying that they have a clinical diagnosis from a medical professional is a confounding factor. Given these limitation  I would urge the authors to try and evaluate the children and then match them by cognitive level. If this is not a possibility they should clearly mention this in the limitations section of the manuscript. 

Author Response

Dear Reviewer 1;

Thank you very much for your thoughtful review of our manuscript. Please find following our point-by-point responses to your comments. We believe that the revised version of the manuscript is improved, as a result of your input.

Sincerely,

JD, JE, JH and RH

Reviewer 1 Comments;

The purpose of this study was to describe gait differences between typically developing children and children diagnosed with autism spectrum disorders (ASD). 

We would like to thank the reviewer for taking the time to review our manuscript.

The manuscript is very well written and the results are clearly presented and explained. However, I have a significant methodological issue. You did pair your groups by age and gender. However, you should have paired them by cognitive level too. Differences in cognitive levels could truly influence your results. Even in posture studies, this has been the case.

Thank you for your positive feedback. We acknowledge that matching participants according to cognitive level, predominantly by evaluating the intelligence quotient, is common in the literature. However, we chose not to pair by cognitive level because of the documented data indicating that an individuals’ intellectual ability does not necessarily reflect motor ability (Manjiviona & Prior, 1995). We have revised the text to include this information. Specifically, we added the following text to the first paragraph of the methods section: “We chose to age- and gender-match these pairs independent of cognitive level because deficits in motor ability in high-functioning children with ASD are not a reflection of cognitive ability at the individual level (Manjiviona & Prior, 1995).”

In addition, you don't describe in detail your group with ASD. Where they high functioning children? or did they vary across the spectrum.

We recognize that ASD is a highly heterogenous population and speak to this issue.  We did not attempt to stratify or narrow the sample population in order to reflect the heterogeneity encountered by service providers.  Further, since we did not attempt to correlate ASD severity to severity of gait differences, we did not feel it was necessary to determine the severity level of each child. We have revised the text to explain that each child with ASD in this study was able to follow directions and tolerate marker placement and other study procedures, without classifying the functional levels of individual participants.  Specifically, we added the following to the first paragraph of the methods section: “…..children with ASD were required to have a clinical diagnosis of ASD from a medical professional that was verbally confirmed by each child’s parent. We did attempt to control or classify the level of ASD severity, as we did not aim to correlate functional level to the severity of gait differences between children with ASD peers with TD.”

 Also the fact that you did not verify diagnoss, actually testing the children and relying that they have a clinical diagnosis from a medical professional is a confounding factor. Given these limitation  I would urge the authors to try and evaluate the children and then match them by cognitive level. If this is not a possibility they should clearly mention this in the limitations section of the manuscript. 

We understand your concern that children’s diagnoses were not independently verified by the study team.  However, this was not possible due to resource and logistic limitations. We have revised the related text in the methods section to more clearly reflect this limitation: “To participate, children with ASD were required to present proof of clinical diagnosis of the disorder from a medical professional that was verbally confirmed by each child’s parent.” Additionally, we have included the following statement in the limitations section: “A limitation to the current study was the inability to independently verify the ASD diagnoses. We addressed this limitation by requiring medical proof of diagnosis and asking the parent(s) of each child with ASD to verbally confirm their child’s diagnosis.”

Reviewer 2 Report

The authors studies 20 participants (10 TD, 10 ASD) in 20 walking trials at self-selected pace. 19 markers were predisposed. Groups were matched for age and gender, but no information was provided for IQ. This is a major limitation given that general developmental delay may impact both on the cognitive and the motor domain. The authors found numerous significant differences in the vGRF, AP GRF, and sagittal plane for hip, knee, and ankle. However, I found these results hard to follow. The authors in the Discussion wrote that ASD participants use distinct motor strategies compared to controls; however the paper seems to furnish more a description of the results than a (potential) explanation/discussion of them.

Even if potential gait anomalies in ASD could be an interesting topic, I would suggest to more deeply characterize and describe the scientific value of the study. At this point, it seems to be more a descriptive report than a scientific paper.

GENERAL POINTS:

* The authors seem to sustain that motor difficulties in ASD have a causal role in their social interaction difficulties. However, recent studies proposed to distinguish between fine/gross motor difficulties  in ASD (i.e. without a direct link to social functioning) and motor cognition difficulties in ASD (i.e. with a direct link to social functioning, for example in motor based action understanding, motor resonance mechanisms, mirror neurons, etc). I would suggest to briefly implement this part in the Discussion.

* It may be useful for readers to have a graphic illustration of marker-set.

* the results are hard to follow. It is not intuitive for not-specialist readers. Graphs are hard to intepret, and they are without clear legends.

* Results reported in Fig.2 seem very hard to interpret, in light of differences in right and left joint angles

* Methodology and statistical analysis are not clearly justified

MAJOR CRITICISMS:

* what about IQ in the ASD group and controls ? Are they matched for cognitive abilities ? if they are not matched for IQ, it may be a problem given that motor delay may be - at least partially - influenced by the general intellectual delay (that include cognitive, but also motor, skills). In other words: if  (some or all) ASD participants have intellectual delay, this may biased the results also in the specific motor task.

* Were clinical diagnosis of ASD confirmed with gold standard instruments (e.g. ADOS) ?

* what about the scientific hypothesis at the basis of this study ? Did authors simply hypothesize that ASD participants would be different compared with controls ? Why ? It is true that the authors recalled the literature on motor difficulties in ASD, but it should be more attentively framed.

SPECIFIC POINTS:

line 21-22 : "Children with ASD exhibited unique differences throughout the gait cycle, which supports current literature on the heterogeneity of the disorder". Why unique differences should support the heterogeneity of ASD?

line 24-25: "Thus, individuals may benefit from therapeutic movement interventions that 24 follow precision medicine guidelines, given the unique movement differences observed"

I do not see what does it means.

lines 32-33: the estimated prevalence reported by authors is very high (1/45; 1/68). They reported also the reference, but I suggest to refer also to others estimate that report lower rate.

lines 42-44: "Delayed diagnoses may be related to the lack of developmental skills at such a young age. Contemporary research suggests that ASD is not solely a mental health disorder, and that it may originate as a movement disorder"

The first phrase is not clear. What did they intend with "developmental skills" ?.

The idea that ASD may "originate" as a movement disorder seems to be too strong. Maybe, it may be that motor anomalies play a role in the pathophysiology of ASD.

lines 69-70: "However, no study has examined lower extremity gait parameters 69 performed across a complete gait cycle". Are the authors sure about it ??

lines: 83-88: Are these real predictions? Please, argue why and how you made these predictions. In addition, the last phrase is not clear. If the observed differences between groups should be unique in terms of directionality, why do you refer to the distinct neurological (?) manifestatios of the ASD spectrum ? Or do you simply mean that ASD are different compared with controls ?

Author Response

Dear Reviewer 3;

Thank you very much for your thoughtful review of our manuscript. Please find following our point-by-point responses to your comments. We believe that the revised version of the manuscript is improved, as a result of your input.

Sincerely,

JD, JE, JH and RH

Reviewer 3 Comments:

GENERAL POINTS:

* The authors seem to sustain that motor difficulties in ASD have a causal role in their social interaction difficulties. However, recent studies proposed to distinguish between fine/gross motor difficulties in ASD (i.e. without a direct link to social functioning) and motor cognition difficulties in ASD (i.e. with a direct link to social functioning, for example in motor based action understanding, motor resonance mechanisms, mirror neurons, etc). I would suggest to briefly implement this part in the Discussion. 

We would like to thank the reviewer for taking the time to review our manuscript. We do not mean to suggest that a causal relationship has been established.  However, there is an emerging body of evidence suggesting that delayed or dysfunctional motor development may in fact influence development in other domains, including communication and cognition.  This is the case we have tried to make.

That said, we believe this comment was strongly related to a particular sentence in the introduction section that originally stated that ASD “may instead originate as a movement disorder,” which could give the wrong impression to readers. We have revised this sentence in the introduction to state the following: “Contemporary research indicates that ASD is not solely a mental health disorder, and that it may also present clinically as a movement disorder (citations) characterized by different manifestations across body systems and developmental domains.” Additionally, in the first paragraph of the discussion section we attempted to explicitly state the lack of a causal relationship. Specifically, we added the following text: “It should be noted that although these children with ASD exhibited gross motor differences compared to their age- and gender-matched peers with TD, a causal relationship between motor dysfunction and impaired behavioral and communication skills has yet to be established.”

* It may be useful for readers to have a graphic illustration of marker-set.

We appreciate your comment and understand this may be helpful for some readers. It is our opinion that since our methods follows standard procedures in gait analysis that this would add little to the context of the experiment.

* the results are hard to follow. It is not intuitive for not-specialist readers. Graphs are hard to intepret, and they are without clear legends.

Thank you for your observation. We have gone back through the results several times and find that they are laid out in a clear, concise fashion, separated by category of measures (kinetic followed by kinematic). Thus, we respectfully disagree with your observation. To your point, however, we offer that it is the interpretation of the results that may be more challenging for the non-specialist reader (not unexpected). To address this interpretation of your comment, we have made several changes to the graphic and tabular presentation of the results, as well as to the Discussion section. 

* Results reported in Fig.2 seem very hard to interpret, in light of differences in right and left joint angles

>>>Figure 2 provides the average number of data point differences across the complete gait cycle (expressed as a percentage of the gait cycle sub-phase). Data corresponding to the average percentage of differences across matched pairs are presented for each joint and for each limb. We have revised the figure legend to better explain the figure. The figure legend now reads, “Figure 2. Mean ± one standard deviation of the number of significant sagittal plane joint angle differences (expressed as a percentage of the gait cycle sub-phase) between TD and ASD children averaged across the 10 matched pairs for the left (top) and right (bottom) hip, knee, and ankle joints.” Additionally, we have revised the y-axis caption to state: “Percentage (%) of Differences”.

* Methodology and statistical analysis are not clearly justified

To more clearly justify our methodology and statistical analyses we have added the following sentence: “To evaluate the complete gait cycle, we used a contemporary data analysis procedure (Dufek et al. 2016) that statistically compares each of the 101 data points across the gait cycle per variable. We have further explained the use this statistical test by later stating: “… we performed this procedure on a matched-pair basis to account for any within subject variability that might have influenced movement pattern differences between children with ASD and children with TD.”

MAJOR CRITICISMS:

* what about IQ in the ASD group and controls ? Are they matched for cognitive abilities ? if they are not matched for IQ, it may be a problem given that motor delay may be - at least partially - influenced by the general intellectual delay (that include cognitive, but also motor, skills). In other words: if  (some or all) ASD participants have intellectual delay, this may biased the results also in the specific motor task.

Thank you for this comment. We acknowledge that matching participants according to cognitive level, predominantly by evaluating the intelligence quotient, is common in the literature. However, we did not pair by cognitive level because of the documented data indicating that an individuals’ intellectual ability does not reflect motor ability (Manjiviona & Prior, 1995). We have revised the text to include this information. Specifically, we added the following text to the first paragraph of the methods section: “We chose to age- and gender-match these pairs independent of cognitive level because deficits in motor ability in children with less severe ASD do not reliably correlate with cognitive ability at the individual level (Manjiviona & Prior, 1995).”

* Were clinical diagnosis of ASD confirmed with gold standard instruments (e.g. ADOS) ? 

Thank you for this comment. We acknowledge concerns  regarding independent verification of each child’s diagnosis. Unfortunately, this was not possible given available resources and logistics.. In light of this limitation, we have revised the related text in the methods section to state: “To participate, children with ASD were required to present proof of a clinical diagnosis of the disorder from a medical professional that was verbally confirmed by each child’s parent.” Additionally, we have included the following statement in the limitations section: “A limitation of the current study was the inability to independently verify each child’s ASD diagnosis. We attempted to address this limitation by asking the parent(s) of each child with ASD to verbally confirm their child’s diagnosis.”

* what about the scientific hypothesis at the basis of this study ? Did authors simply hypothesize that ASD participants would be different compared with controls ? Why ? It is true that the authors recalled the literature on motor difficulties in ASD, but it should be more attentively framed.\

Thank you for this comment. We have addressed these concerns/questions below in the reviewer’s section of “specific points.”

SPECIFIC POINTS:

line 21-22 : "Children with ASD exhibited unique differences throughout the gait cycle, which supports current literature on the heterogeneity of the disorder". Why unique differences should support the heterogeneity of ASD?

Unique differences, as observed and described in this study, indicate that each child with ASD exhibits different walking mechanics compared to their TD control, though the direction of the differences (lesser vs. greater joint flexion for example) is not consistent across the matched pairs. Thus, no identifiable gait pattern or set of characteristics can be considered as a step toward diagnosis. ASD is heterogeneous in that no two individuals with ASD function in the same way (Brincker & Torres, 2013).  In our study, no two individuals with ASD exhibited identical, or even similar, movement differences compared to their TD controls. The current results provide support for the recent literature stating that ASD is a disorder characterized by heterogeneous manifestations (Rinehart et al. 2006; Weiss et al. 2013).

line 24-25: "Thus, individuals may benefit from therapeutic movement interventions that 24 follow precision medicine guidelines, given the unique movement differences observed"

I do not see what does it means.

This statement reflects the growing acknowledgement that the “average patient” and “one-size-fits-all” approaches of most medical treatments is successful only for some patients but not for others. A major goal for this project was to present data in support of the “precision medicine” approach, which takes into account each person’s physiological characteristics, environments, and lifestyles. We believe that ASD treatments must follow precision medicine guidelines for personalized care to provide the most appropriate rehabilitation treatments. To clarify the original statement, we revised the text to explicitly state that precision medicine emphasizes individual characteristics during treatment. Specifically, we added, “Thus, individuals may benefit from therapeutic movement interventions that follow precision medicine guidelines by accounting for individual characteristics, given the unique movement differences observed.”          

lines 32-33: the estimated prevalence reported by authors is very high (1/45; 1/68). They reported also the reference, but I suggest to refer also to others estimate that report lower rate.

Our goal with this statement is to show that the prevalence of ASD has consistently risen in recent years, and we feel as though minor discrepancies between the cited documentation and other documentation do not alter the accuracy of what we have reported. That said, we have revised the text to read “The prevalence of ASD has continually risen in recent years with current prevalence estimations ranging from one in 45 to one in 68” to avoid any confusion.

lines 42-44: "Delayed diagnoses may be related to the lack of developmental skills at such a young age. Contemporary research suggests that ASD is not solely a mental health disorder, and that it may originate as a movement disorder"

The first phrase is not clear. What did they intend with "developmental skills" ?. 

We agree that the first phrase was not clear in the manuscript as originally submitted. Our intent was to explain that, chronologically, social and communication skills develop later than movements. The work of Kuhl suggests that motor activity associated with expressive language is present prior to other detectable verbal or social responses (Kuhl, Ramirez, Bosseler, Lin & Imada, 2014). We have revised this sentence to state, “Delayed diagnoses may be related to the small repertoire of observable social and communication skills at such a young age.” We believe this revision more directly explains the difference in the age of typical diagnoses to the suggestions of Teitelbaum (1998), which posit that movement-based diagnoses can be made during infancy.

The idea that ASD may "originate" as a movement disorder seems to be too strong. Maybe, it may be that motor anomalies play a role in the pathophysiology of ASD.

Thank you for this comment. While the term “originate” may be a bit too strong webelieve that motor anomalies play a role in ASD, as suggested in a growing body of literature (Lobo et al, 2013). Our intent was to make an empirically supported statement that ASD can present clinically as a movement disorder in addition to a mental health disorder, and that early motor delay or dysfunction may contribute to social communicative dysfunctions. We have addressed this throughout the manuscript to replace all references to ASD originating as a movement disorder to now state that ASD can present clinically as a movement disorder.

lines 69-70: "However, no study has examined lower extremity gait parameters 69 performed across a complete gait cycle". Are the authors sure about it ??
To the best of the authors’ knowledge, no study has mechanically evaluated lower extremity kinetic and kinematic gait parameters across a complete gait cycle in this population as most have instead opted for discrete point analyses or have examined range of motion parameters (Calhoun, Longworth, & Chester, 2011; Chester & Calhoun, 2012; Hallet et al. 1993; Rinehart et al. 2006). We have revised this sentence to state, “However, to the authors’ knowledge, no study has examined lower extremity gait parameters performed across a complete gait cycle by using a point-to-point analysis.”

lines: 83-88: Are these real predictions? Please, argue why and how you made these predictions. In addition, the last phrase is not clear. If the observed differences between groups should be unique in terms of directionality, why do you refer to the distinct neurological (?) manifestatios of the ASD spectrum ? Or do you simply mean that ASD are different compared with controls ?

Yes, based on the body of literature, we hypothesized that children with ASD would exhibit different movement characteristics than their age- and gender-matched TD peers. Further, we also hypothesized that the differences were expected to be heterogeneous (different directionality from matched-pair to matched-pair) based on the understanding of ASD as a complex, heterogeneous, neurodevelopmental disorder, akin to cerebral palsy. We refer to distinct neurological manifestations because the documented heterogeneity of neurological manifestations has lead researchers to conclude that ASD affects each child’s neurological system differently (Rinehart et al. 2006; Kindregan, Gallagher, & Gormley, 2015). In light of those conclusions, we anticipated a similar outcome with respect to each child’s motor system.

Our reasoning for these hypotheses is supported by the literature, however, we may not have stated our expected outcomes in a manner that clearly communicated them. To address this issue, we have revised our hypotheses to state that, “Due to the distinct neurological manifestations of ASD (citations), we expected similar observations with respect to motor function. As such, we hypothesized that statistically significant differences, unique in terms of directionality of differences (increased vs. decreased flexion/force), would be revealed across the gait cycle for each matched-pair with respect to hip, knee, and ankle joint angles and vertical and anterior-posterior ground reaction forces (vGRF and AP GRF, respectively).”

Reviewer 3 Report

This is a good paper on a topic that is important. 

To make it a more relevant paper the authors ought to relate their findings to the work done by many others who have done empirical as well as foundational work in this field. That is, the present paper will have much more value if the authors can connect their work to other work in movement and autism, particularly the work of Elizabeth Torres. I note that this authors cite from a Frontiers in Integrative Neuroscience issues she co-edited but do not refer to any of the work in that issue which might tell an interested reader how the present paper might contribute to other work which attempts to understand what autism is. 

Author Response

Dear Reviewer 2;

Thank you very much for your thoughtful review of our manuscript. Please find following our point-by-point responses to your comments. We believe that the revised version of the manuscript is improved, as a result of your input.

Sincerely,

JD, JE, JH and RH

Reviewer 2 Comments:

This is a good paper on a topic that is important.

We would like to thank the reviewer for taking the time to review our manuscript and acknowledging our efforts to address an important topic.

To make it a more relevant paper the authors ought to relate their findings to the work done by many others who have done empirical as well as foundational work in this field. That is, the present paper will have much more value if the authors can connect their work to other work in movement and autism, particularly the work of Elizabeth Torres. I note that this authors cite from a Frontiers in Integrative Neuroscience issues she co-edited but do not refer to any of the work in that issue which might tell an interested reader how the present paper might contribute to other work which attempts to understand what autism is

Thank you for this comment. Although we are confident that we connected the current work to other work emphasizing movement in ASD, it is true that additional information will strengthen our conclusions. Specifically, we have revised our manuscript to include the work of Elizabeth Torres, as suggested. We revised the end of first paragraph of the introduction section to state, “Unfortunately, the behavioral emphasis during subjective clinical observations and treatments has done little to bring positive changes to individuals affected by ASD (Torres & Donnellan, 2015).” We have also included additional literature in the discussion section in an attempt to better connect movement and ASD for our readers. Specifically, we revised the end of the first discussion paragraph to state, “This finding is consistent with the literature (citations) documenting the heterogeneous nature of ASD, and supports the fact that no two individuals with ASD are the same (Brincker & Torres, 2013).” We also added additional information later in the discussion. Specifically, we revised the text to state, “It is likely that the distinct strategies observed reflect inconsistent or unpredictable movement responses to different sources of peripheral noise (Brincker & Torres, 2013) and different sensory capabilities (Tomchek & Dunn, 2007) affecting proprioception when the foot contacts the ground.”

Round 2

Reviewer 1 Report

I would like to thank the authors for the detailed response to our comments. However, with regards to matching the children by cognitive level, the authors use a very old reference, that cognitive and motor tasks are not associated. This is not the case as there are multiple studies that suggest the opposite. I would suggest to add this as a limitation of their study, so that the study can be accepted. 

Author Response

We have added text to the revised manuscript which addresses your comment. Specifically, in the limitations we have added the following text:

Although we did not pair based on cognitive level according to Manjiviona and Prior [33], we acknowledge that this may be a limitation as other studies suggest cognitive level is a necessary control in this population. 

Thank you for your comment.

Reviewer 2 Report

Even if I recognize authors' effort in order to address my previous concerns, I remain skeptical about the major criticisms raised in my previous report.

Beyond minor points (some of them have been well addressed, other remain to some extent problematic), I think that in thiw work remain the major criticisms concerning the absence of clinical characterization of participants and the (potential) presence of cognitive delay in some (or all, we cannot know it from the paper) patients.

In my opinion, authors' reply (i.e. "However, we did not pair by cognitive level because of the documented data indicating that an individuals’ intellectual ability does not reflect motor ability (Manjiviona & Prior, 1995)" ) cannot be accepted. In fact, in the litterature is very clear that intellectual delay is strongly related (also to) to motor difficulties. Thus, the groups (TD and ASD) in this work are neither matched nor characterized. I am wondering how a clinical study (i.e. a study with patients) may be accepted when the patients are not described.  It is also unacceptable in my opinion the fact that patients had not a clear diagnosis (authors even referred to parent's report).  I do not think that demanding clinical characterization (e.g. IQ) or clear diagnosis (e.g. ADOS) is an excessive pedantry. In my opinion is crucial in clinical studies given that you are comparing a typical and a patient group (but you are not really sure about the clinical feature of your clinical sample).

Author Response

Thank you for your comments.

We have added text to the limitations section to address your concern:

Although we did not pair based on cognitive level according to Manjiviona and Prior [33], we acknowledge that this may be a limitation as other studies suggest cognitive level is a necessary control in this population.